# Rapid progression and short-term mortality during the January 2023 to July 2024 Cholera outbreaks in Zambia: A retrospective facility-based study

Deborah Tembo[1], Nedah Chikonde Musonda[2], Miyanda Simwaka[3], Chipo Nkwemu[4,5], Samson Shumba [5]*

1 Department of Emergency Preparedness and Response, Zambia National Public Health Institute, Lusaka, Zambia, 2 Copper Rose Zambia, Lusaka, Zambia, 3 Ministry of Health, Lusaka, Zambia, 4 Centre for Infectious Disease Research in Zambia, Lusaka, Zambia, 5 Department of Epidemiology and Biostatistics, School of Public Health, University of Zambia Ridgway Campus, Lusaka, Zambia

* samsonshumba1@gmail.com

## Abstract

Cholera remains one of the most rapidly fatal infectious diseases globally, with untreated cases leading to death within hours due to severe dehydration. Despite being preventable and treatable, it continues to pose a significant public health threat, especially in low and middle-income countries where water, sanitation, and hygiene (WASH) services are inadequate. This study aimed at investigating the rapid progression and short-term mortality during the January 2023 to July 2024 Cholera Outbreaks in Zambia. We conducted a retrospective cross-sectional study using national cholera surveillance data from Zambia's January 2023 to July 2024 outbreak to examine delays in care and associated outcomes. All suspected and confirmed cases with complete records on symptom onset, admission, and outcome were included. Key time intervals were calculated and analyzed using descriptive statistics and non-parametric tests, with subgroup comparisons by age, sex, and outcome status. Ethical approval was obtained, all data were anonymized, and analyses were performed using Stata version 14.2. A total of 3,655 cases were analyzed, of whom 52.2% were male and 37.8% were under 15 years. Nearly all patients (99%) were admitted as inpatients, and the overall mortality rate was 4.2%. The median time from symptom onset to death was 3 days, underscoring the rapid progression of the disease. Of the 3,655 cholera patients enrolled in the study, 2,162 patients were included in the survival analysis, the death rate was 0.85 per 100 person-days, with higher rates observed among males (1.05) and adults aged 50 years and above (2.71). Confirmed cases experienced the highest mortality, and survival probabilities were significantly lower among males (p = 0.0498), older adults (<0.0001), and confirmed cholera cases (<0.0001). Many cholera deaths in this outbreak happened within just a few days of admission, showing the need for quick assessment and

**Data availability statement:** The data supporting the findings of this study are available from the Zambia National Public Health Institute (ZNPHI) (https://www.znphi.org.zm ) and the Ministry of Health, Lusaka, Zambia (https://www.moh.gov.zm ). While the dataset has been attached to this submission, access is subject to approval from ZNPHI and the Ministry of Health, who remain the authorized custodians of the data. Requests for data access may be directed to ZNPHI via email at info@znphi.co.zm or by telephone at (+260) 211 269 432.

**Funding:** The author(s) received no specific funding for this work.

**Competing interests:** The authors have declared that no competing interests exist.

urgent fluid treatment as soon as patients arrive. Older adults and men faced a higher risk of dying, which calls for triage protocols that pay closer attention to these groups. Laboratory confirmation was limited, with very few patients receiving rapid diagnostic tests, revealing a serious gap in surveillance and case management. Strengthening early warning systems, improving access to rapid tests, and focusing on high-risk groups are key steps to lowering deaths in future outbreaks.

## Background

Cholera remains one of the most rapidly fatal infectious diseases around the world, capable of causing death within hours due to severe fluid loss if untreated [1,2]. It is caused by ingestion of *Vibrio cholerae*, a bacterium producing enterotoxins that disrupt intestinal ion transport, resulting in profuse watery diarrhea and vomiting [3,4]. Without prompt rehydration, severe dehydration leads to the majority of cholera-related deaths [5]. Transmission of this disease thrives in areas with poor sanitation and unsafe water, making cholera endemic in many low and middle income countries [6]. Outbreaks are associated with natural disasters or occur in overcrowded urban settings with inadequate water and sanitation infrastructure [7,8].

Globally, cholera continues to impose a heavy burden with about 1.3 to 4 million cases and 21,000–143,000 deaths annually estimated by the World Heath Organization (WHO) [9–11]. In 2022, 44 countries reported a 25% increase in outbreaks from the previous year with sub-Saharan Africa disproportionately affected, accounting for over 30% of cases and more than 80% of cholera deaths worldwide [5,12]. Between 2000 and 2015, this region suffered 83% of global cholera mortality, driven by fragile health systems, high population density, and inadequate water, sanitation, and hygiene (WASH) services [13,14].

Zambia's latest cholera resurgence emerged in October 2023, with Lusaka Province at its epicenter. Within three months, the outbreak had spread to nine provinces, resulting in more than 10,887 reported cases and 432 deaths by January 2024, with Lusaka, Central, and Eastern provinces bearing the greatest burden. Cholera has been endemic in Zambia since 1977, with outbreaks recurring every three to five years [15]. The most recent resurgence began in October 2023, originating in Lusaka Province, and by mid-2024 had resulted in over 23,000 reported cases and 740 deaths [16]. Lusaka, marked by dense and unplanned settlements, has historically been the epicenter, accounting for 73% of national cases between 2001 and 2010 [17,18]. The disease disproportionately affects children, with 48% of cases occurring in those under 15 years and 32% in children under five, while males consistently constitute the majority of both cases and fatalities [19]. Although Zambia's National Cholera Elimination Plan set a target for elimination by 2025, the scale of the current outbreak renders this goal unlikely in the near term [20].

Timely treatment is critical in cholera management; without intervention, the disease can have a Case Fatality Rate (CFR) exceeding 50%, whereas prompt oral or intravenous rehydration can reduce case fatality to below 1% [2,21]. In Zambia,

delays in seeking and receiving care remain a major concern, as evidenced by the substantial number of deaths occurring outside health facilities [17]. By early 2024, the national CFR was 3.5%, markedly higher than the 1.8% recorded within healthcare settings [22,23]. Barriers to timely care include long distances to health facilities, transportation challenges, financial constraints, reliance on home remedies, and cultural factors influencing treatment decisions [24,25], with these obstacles being especially pronounced in peri-urban communities [24].

Although oral cholera vaccines (OCVs) have been widely used, their protection wanes after two to three years [26], especially in young children, and limited access to Oral Rehydration Points (ORPs) contributes to ongoing community deaths. These challenges highlight the urgent need for data-driven analysis of delays in care from symptom onset to treatment and outcome. This study aimed at investigating the rapid progression and short-term mortality during the January 2023 to July 2024 Cholera Outbreaks in Zambia.

## Conceptual framework

The study used a detailed conceptual framework to analyze delays in cholera case management and factors influencing health outcomes. It considered individual, community, and system-level aspects across demographic, clinical, structural, behavioral, and environmental domains. Key variables included patient demographics, clinical signs, treatment, and geographic factors like distance to health facilities. Outcomes measured are time from symptom onset to resolution, survival status, and place of death as an indicator of delayed care (See Fig 1).

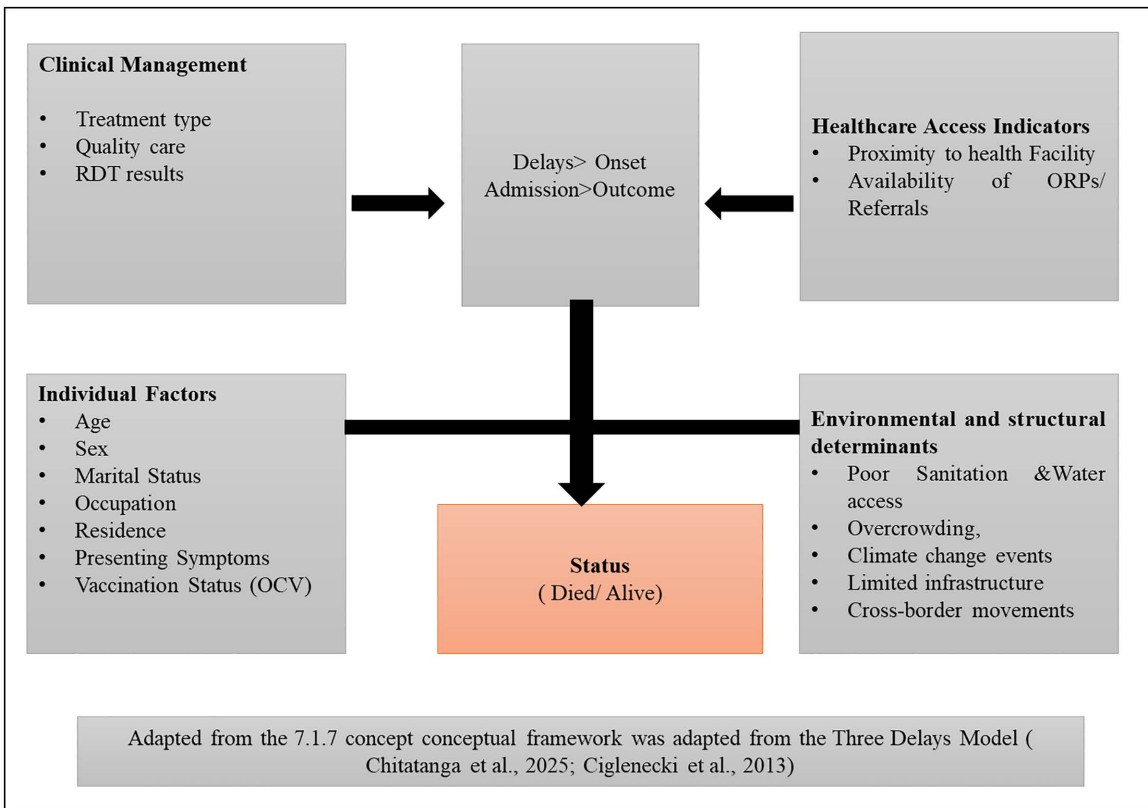

**Fig 1. Integrated conceptual framework for analyzing delays and determinants of cholera case management and outcomes in Zambia.**

## Methodology

### Study design, setting and population

This study utilized a retrospective cross-sectional design, drawing on national surveillance data from the 2023–2024 cholera outbreaks in Zambia. Data were obtained from health facilities nationwide where suspected cholera patients were admitted and managed. The analysis examined delays between symptom onset, hospital admission, and completion of the Case Outcome Investigation (COI) process. All suspected and confirmed cholera cases reported between 21 January 2023 and 7th July 2024 were included, encompassing patients of all ages and sexes who sought care at health facilities and had complete records for the variables of interest.

### Data sources and collection

Data were obtained from the national cholera surveillance system managed by the Ministry of Health, accessed on 8 August 2024. This included patient-level records obtained from facility registers and the electronic cholera case reporting tools. Key variables included date of symptom onset, date of admission, and date of case outcome (recovery or death), age, and sex, province, and outcome status.

### Variables and operational definitions

Key demographic and clinical characteristics were included to assess delays in cholera case management. Age was grouped into predefined categories to capture age-related differences.. Patient admission status, vital status at reporting, including laboratory testing performed before notification in the surveillance system, provided context on case severity and diagnosis. Admission status indicated the level of care required, vital status captured immediate survival outcomes, and laboratory testing prior to notification helped distinguish confirmed cases from suspected or probable cases. Outcomes from the Case Investigation Outcome (COI) process informed final patient status and case classification. The main focus was on quantifying time intervals from symptom onset to key milestones in care and case closure, see Table 1 below.

**Table 1. Key variables with stata-compatible names, measurements, and operational definitions.**

| Variable Name | Measurement | Operational Definition |
|---|---|---|
| Age group | Categorical (7 levels) | Age grouped as: 0–14, 15–24, 25–34, 35–49 and 50+years. |
| Sex | Categorical (Male, Female) | Biological sex of the patient as recorded in facility register. |
| Patient category | Categorical (Inpatient, Outpatient) | Classification of patient on admission. |
| Status at report | Categorical (Alive, Dead) | Patient's vital status at the time of case notification/report. |
| Rapid test done | Binary (Yes/No) | Whether a rapid diagnostic test was done before notifying the health authorities. |
| COI vital status | Categorical (Alive, Dead) | Final outcome of the patient during the Case Investigation Outcome (COI) process. |
| COI exit status | Categorical (Recovered, Not a Case, Unknown) | Final categorization of case after investigation and care. |
| Days onset to discharge | Continuous (Days) | Number of days from symptom onset to patient discharge. |
| Days onset to COI closure | Continuous (Days) | Number of days between symptom onset and COI case closure. |

## Sample size

This study is based on a national surveillance dataset capturing all suspected and confirmed cholera cases reported during the 2023–2024 country-wide outbreak. By including the entire population of eligible cases with complete timing data, the study adopts a census approach, eliminating sampling variability and ensuring robust, highly generalizable findings on delays in cholera case management.

## Data management and analysis

Data cleaning and management were conducted using Microsoft Excel and Stata 14.2. Records with missing or inconsistent dates were excluded from the analysis. Time intervals were computed between: admission to patient outcome, admission and COI conclusion, symptom onset and COI conclusion. Descriptive statistics (means, medians, interquartile ranges) were used to summarize time delays. Stratified analysis was done by age group, sex, province, and outcome (recovered vs. died). Where appropriate, non-parametric tests (Wilcoxon rank-sum) were used to assess differences in delays across groups. Additionally, Kaplan-Meier survival curves were generated to visualize and compare the time-to-event distributions, such as time from admission to outcome, across different strata.

## Ethical considerations

Ethical approval for this study was obtained from the University of Zambia Biomedical Research Ethics Committee (**Ref No. 5066−2024**). Additional authorization to access and use anonymized national cholera surveillance data was granted by the Ministry of Health and the Zambia National Public Health Institute. The dataset contained no personal identifiers, and all data were handled securely and used solely for research purposes, in accordance with national and international ethical standards.

## Results

### Descriptive characteristics of patients presenting with cholera

A total of 3,655 patients presented with suspected or confirmed cholera. The largest proportion (37.8%) were aged 0–14 years, followed by those aged 25–34 years (20.2%), 35–49 years (17.3%), and 15–24 years (16.2%). Patients aged 50 years and above accounted for 8.4% of the total. In terms of sex, 52.2% of the patients were male and 47.8% were female. At the end of follow-up, 95.8% of patients were reported as alive, while 4.2% had died. According to the case outcome investigation (COI) classification, 16.8% were confirmed cholera cases, 67.5% were suspected or probable cases, 13.9% were asymptomatic, and 1.8% were classified as non-cases or other (See Fig 2).

### Diagnostic and admission characteristics of patients presenting with Cholera

Out of 3,655 patients, the vast majority (99.0%) were admitted as inpatients, while only 0.96% were managed as non-inpatients. Rapid diagnostic tests (RDTs) were performed in 11.2% of the cases. However, 56.1% of patients did not undergo RDTs, and the test status was unknown for 32.7%. Among those tested, 83.3% returned positive results, 16.3% tested negative, and 0.4% had an error result. Regarding laboratory testing, serological (ELISA) tests were the most commonly used method, accounting for 76.0% of tests performed. Stool culture was conducted in 21.2% of cases, other cultures in 2.4%, and PCR tests were used in 0.5% of the patients (See Fig 3).

### Time intervals in the clinical course and management of cholera patients

The Table 2 summarizes key time intervals related to the progression and management of cholera cases. The median time from admission to death was 2 days, with most deaths occurring between 1 and 3 days after admission. Similarly, the median time from symptom onset to death was 3 days, with an interquartile range of 2–4 days. The duration from

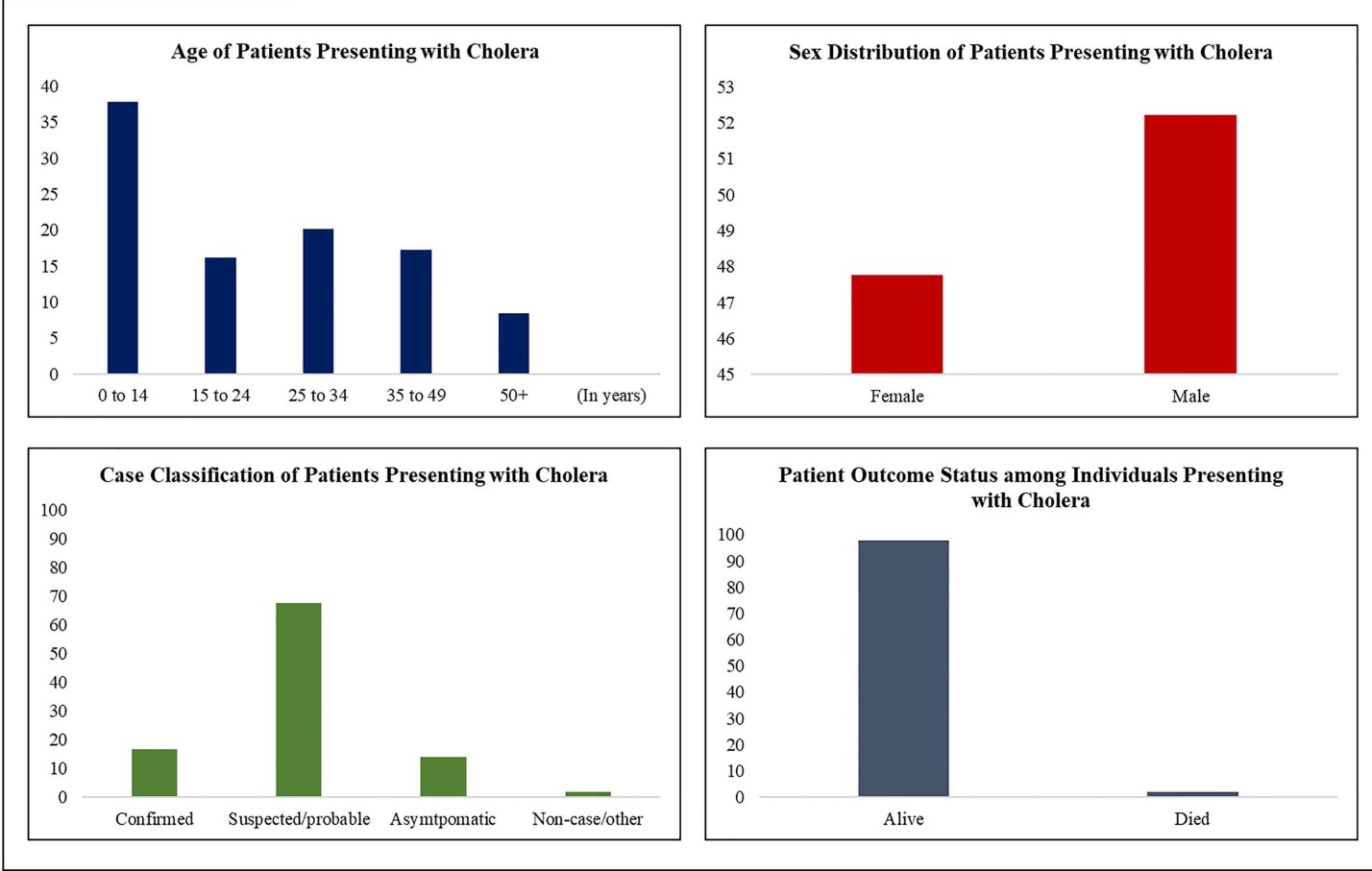

**Fig 2. Age, sex, vital status, and case classification of patients presenting with cholera.**

symptom onset to case outcome investigation (COI) conclusion had a median of 3 days, with most cases concluded between 1 and 4 days. From admission to COI conclusion, the median time was 2 days, ranging from 0 days conclusions up to 3 days. Specimen collection occurred rapidly after symptom onset, with a median of 0 days and an interquartile range of 0–1 day.

Other intervals related to specimen processing were also brief. The median time from admission to specimen collection was 0 days, as was the time from specimen receipt to laboratory dispatch. The interval from specimen collection to receipt had a median of 0 days, with an interquartile range of 0 to 1 day. Finally, the median time from symptom onset to first healthcare contact was 0 days, with most patients seeking care within 2 days of symptom onset.

### Survival outcomes among cholera patients

Of the 3,655 cholera patients enrolled in the study, 2,162 with available data on time from admission to death were included in the survival analysis as shown in Table 3 below. These patients contributed a total of 6,386 person-days of follow-up, yielding an overall incidence rate of death of 0.85 per 100 person-days. Median and other percentile survival times could not be estimated for the overall cohort. By sex, males (n = 1,190) had a higher incidence rate of 1.05 deaths per 100 person-days compared with 0.60 per 100 person-days among females (n = 972). Males also had an estimable

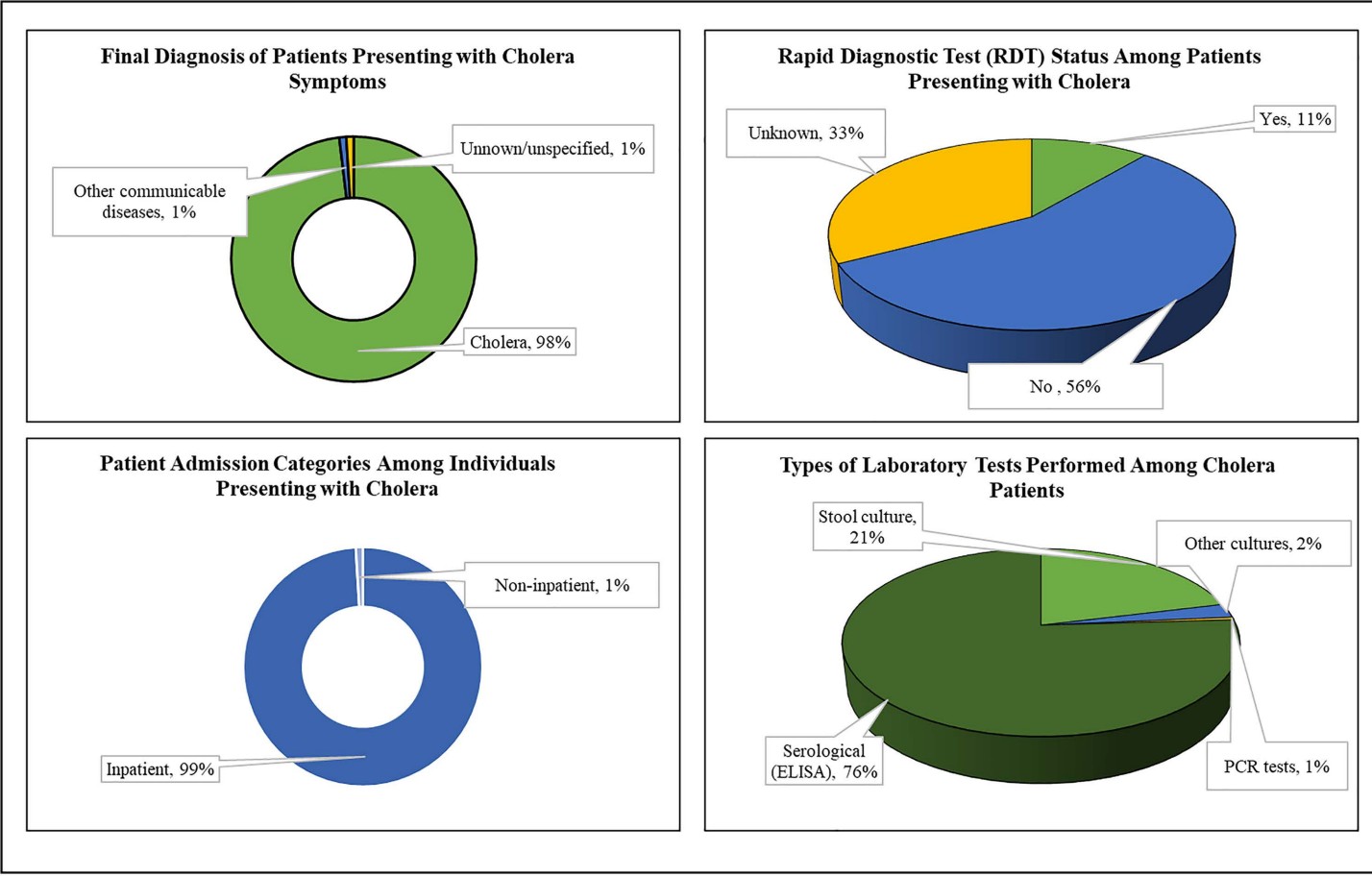

**Fig 3. Admission status, rapid diagnostic testing, and laboratory methods among cholera patients.**

**Table 2. Median and interquartile ranges of key time intervals from symptom onset to outcome among cholera patients.**

| Variables | Medium | Interquartile |
|---|---|---|
| Admission to dead | 2 | 1 - 3 |
| Symptom onset to dead | 3 | 2 - 4 |
| Symptom onset to COI conclusion | 3 | 1 - 4 |
| Admission to COI conclusion | 2 | 0 - 3 |
| Symptom onset to specimen collection | 0 | 0 - 1 |
| Admission to specimen collection | 0 | 0 - 0 |
| Specimen collection to receipt | 0 | 0 - 1 |
| Specimen receipt to lab dispatch | 0 | 0 - 0 |
| Symptom onset to first healthcare | 0 | 0 - 2 |

**Table 3. Survival analysis of cholera patients by sex and age group: incidence rates and percentile survival times.**

| Category | Number of Patients | Time at Risk (Person-Days) | Incidence Rate (Events per 100 Person-Days) | 25th Percentile Survival Time (Days) | Median Survival Time (Days) | 75th Percentile Survival Time (Days) |
|---|---|---|---|---|---|---|
| **Overall** | 2,162 | 6,386 | 0.85 | Not estimable | Not estimable | Not estimable |
| **Sex** | | | | | | |
| Female | 972 | 2,847 | 0.6 | Not estimable | Not estimable | Not estimable |
| Male | 1,190 | 3,539 | 1.05 | 16 | Not estimable | Not estimable |
| **Overall** | 2,162 | 6,386 | 0.85 | Not estimable | Not estimable | Not estimable |
| **Age** | | | | | | |
| 0–14 | 788 | 2,285 | 0.0053 | Not estimable | Not estimable | Not estimable |
| 15–24 | 356 | 927 | 0.0076 | 16 | 16 | 16 |
| 25–34 | 435 | 1,384 | 0.0072 | Not estimable | Not estimable | Not estimable |
| 35–49 | 383 | 1,236 | 0.0081 | Not estimable | Not estimable | Not estimable |
| 50+ | 200 | 554 | 0.0271 | 8 | Not estimable | Not estimable |
| **Case Classification** | | | | | | |
| Confirmed | 312 | 833 | 1.68 | Not estimable | Not estimable | Not estimable |
| Suspect/Probable | 1,336 | 3,929 | 0.61 | 16 | Not estimable | Not estimable |
| Asymptomatic | 380 | 1,167 | 0.6 | Not estimable | Not estimable | Not estimable |
| Non-case/Other | 39 | 89 | 8.99 | Not estimable | Not estimable | Not estimable |

25th-percentile survival time of 16 days, whereas no percentile survival times were estimable for females. The total time at risk was longer among males (3,539 person-days) than females (2,847 person-days).

The age group 50 years and above recorded the highest incidence rate at 2.71 deaths per 100 person-days, with a 25th percentile survival time of 8 days. This contrasts with the 15–24 age group, which had a lower incidence rate of 0.76 and survival percentiles (25th, 50th, and 75th) all estimated at 16 days. The 0–14 age group, which constituted the largest portion of the cohort (788 patients), had the lowest incidence rate of 0.53, with all percentile estimates not available. The 25–34 and 35–49 age groups had similar incidence rates of 0.72 and 0.81 respectively, though no survival percentiles were estimable for either group.

Based on case classification, confirmed cholera cases had an incidence rate of 1.68 deaths per 100 person-days, while suspected/probable cases had a lower rate of 0.61. Asymptomatic cases had a similar rate of 0.60, whereas the non-case/other group recorded the highest rate at 8.99 deaths per 100 person-days, despite being a small sub-group (n = 39).

## Survival probability among cholera patients by sex and age group

The Kaplan-Meier survival analysis shows that most cholera patients survived the follow-up period, with low overall mortality and a non-estimable median survival time. When stratified by sex, males had a steeper decline in survival compared to females. This difference was statistically significant (log-rank test: $p = 0.0498$). Analysis by age group revealed significant differences in survival probabilities ($p < 0.0001$). Patients aged 50 years and above had the lowest survival rates and the most rapid decline in survival, while younger groups, particularly those aged 0–14 and 15–24 years, showed relatively stable and high survival probabilities.

Survival also differed significantly by case classification ($p < 0.0001$). Case classification, confirmed cases demonstrate a higher incidence of death with a sharper decline in survival probability compared to suspect/probable and asymptomatic cases. Suspect/probable cases have a more gradual decrease, while asymptomatic and non-case groups show relatively stable survival curves with minimal declines (See Fig 4).

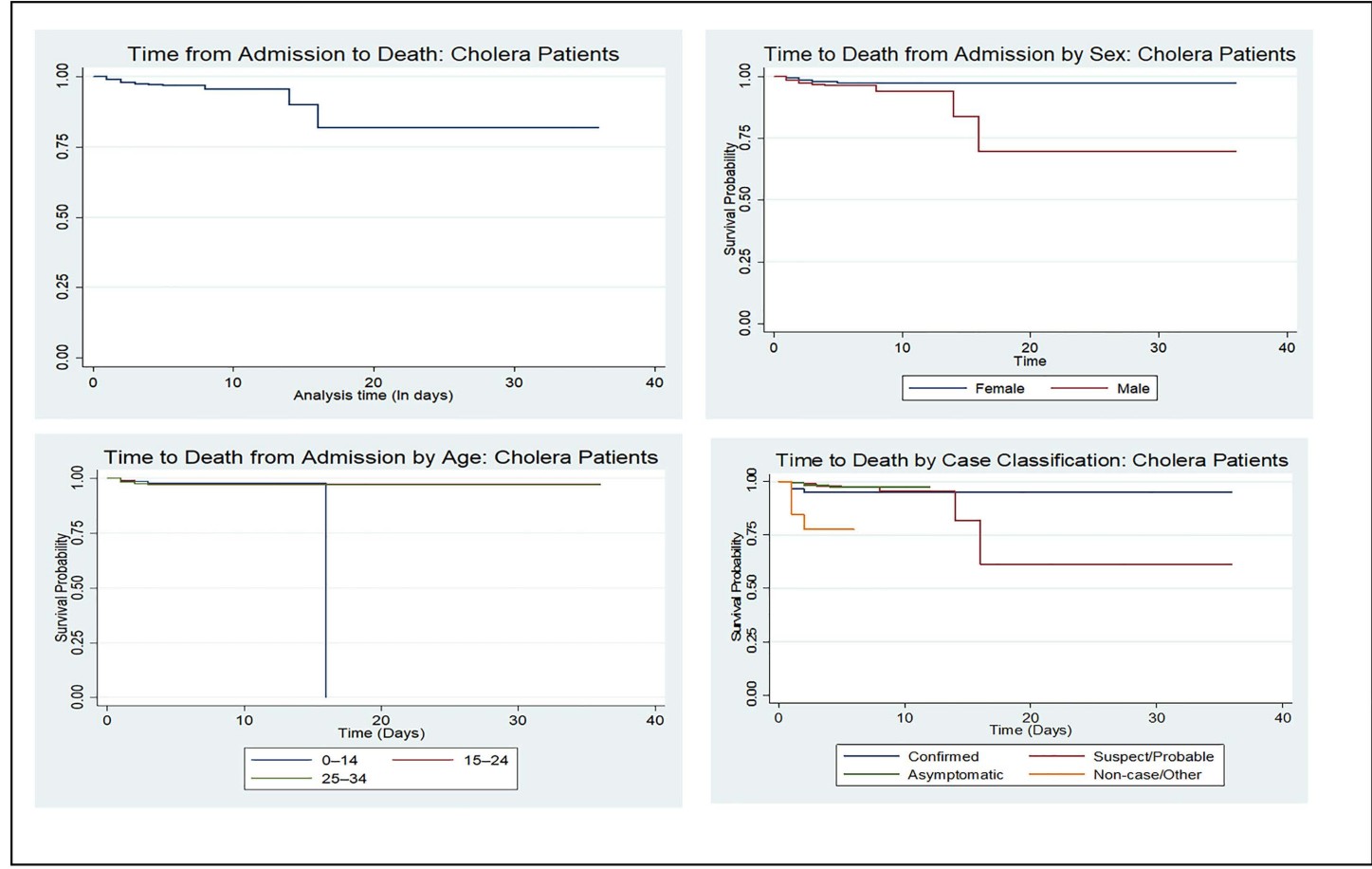

**Fig 4. Kaplan-Meier survival curves for cholera patients stratified by sex, age and case classification.**

### Time to death and survival determinants during the 2023–2024 cholera outbreak in Zambia: a cox regression approach

The Cox proportional hazards model identified several statistically significant predictors of time to death among cholera patients admitted during the 2023–2024 outbreak in Zambia. Sex was a significant predictor of mortality, with males having nearly twice the hazard of death compared to females (aHR = 1.89; 95% CI: 1.03–3.47; $p = 0.039$). Age showed a clear gradient, with individuals aged 50 years and above exhibiting a markedly higher hazard of death (aHR = 6.02; 95% CI: 2.66–13.51; $p < 0.001$) compared to those aged 0–14 years.

Case classification was also associated with survival outcomes. Patients categorized as "Suspect/Probable" (aHR = 0.27; 95% CI: 0.14–0.53; $p < 0.001$) and "Asymptomatic" (aHR = 0.37; 95% CI: 0.14–0.95; $p = 0.040$) had lower hazards of death than confirmed cholera cases. Conversely, those in the "Non-case/Other" category had a significantly higher hazard of death (aHR = 3.74; 95% CI: 1.52–9.17; $p = 0.004$). Geographical variation was observed, with patients from Lusaka Province showing higher mortality hazards compared to those from other provinces (aHR = 9.95; 95% CI: 1.34–71.76; $p = 0.024$) (See Table 4).

**Table 4. Factors associated with time to death among admitted cholera patients in Zambia.**

| Variables | Crude Analysis | | Multivariable Analysis | |
|---|---|---|---|---|
| | cHR (95% CI | P-value | aHR (95% CI | P-value |
| **Sex** | | | | |
| Female | Ref | | Ref | |
| Male | 1.78 (1.00 - 3.15) | 0.051 | 1.89 (1.03 - 3.47) | 0.039*** |
| **Age** | | | | |
| 0–14 | Ref | | Ref | |
| 15–24 | 1.37 (0.54 - 3.49) | 0.506 | 1.35 (0.51 - 3.57) | 0.550 |
| 25–34 | 1.43 (0.62 - 3.30) | 0.407 | 1.35 (0.55 - 3.30) | 0.517 |
| 35–49 | 1.62 (0.70 - 3.74) | 0.263 | 1.62 (0.66 - 3.93) | 0.298 |
| 50+ | 5.18 (2.41 - 11.12) | <0.001 | 6.02 (2.66 - 13.51) | <0.001*** |
| **COI Grouped** | | | | |
| Confirmed | Ref | | Ref | |
| Suspect/Probable | 0.39 (0.20 - 75) | 0.005 | 0.27 (0.14 - 0.53) | <0.001*** |
| Asymptomatic | 0.40 (0.16 - 99) | 0.048 | 0.37 (0.14 - 0.95) | 0.04*** |
| Non-case/other | 5.38 (2.23-12.96) | <0.001 | 3.74 (1.52 - 9.14) | 0.004*** |
| **Province** | | | | |
| Other | Ref | | Ref | |
| Lusaka Province | 6.68 (0.92-48.30) | 0.06 | 9.95 (1.34 - 71.76) | 0.024*** |

**\*\*\*** Statistically significant aHR – Adjusted Hazard Ratio

## Discussion

This study analyzed Zambia's national cholera surveillance data from the 2023–2024 outbreak, encompassing over 3,655 suspected and confirmed cases. Using detailed epidemiological records, we examined key patterns in case fatality, survival times, and demographic risk factors to better understand the outbreak's progression and identify weaknesses in clinical response. Results showed that patients generally sought care very quickly, with a median delay of zero days from symptom onset to first healthcare contact, indicating rapid health-seeking or swift case identification through surveillance. However, despite this early contact, deaths occurred rapidly and frequently. The overall CFR was 4.2%, significantly exceeding the WHO target of less than 1%. Most deaths happened within a short timeframe: the median interval from symptom onset to death was three days, and nearly half of fatalities occurred within 48 hours after admission. These findings highlight the fast progression of cholera and point to critical gaps in timely and effective treatment during the outbreak.

Children under 15 years made up the largest share of cases (37.8%), aligning with national data [16,19], and generally experienced better survival outcomes. In contrast, older adults aged 50 and above had the highest death rates, with an incidence of 2.71 deaths per 100 person-days. Additionally, mortality was notably higher among male patients compared to females. Confirmed cholera cases faced greater mortality risk than suspected cases; however, the highest fatality rates were observed in patients categorized as "non-case/other," a group that likely includes misdiagnosed or severe cases that went undetected. It is also possible that some of these cases met the case definition criteria but were confirmed later. During outbreaks, sample collection often occurs in stages due to logistical challenges [27]. This phased strategy is particularly important in cholera outbreaks, where rapid disease spread demands efficient allocation of diagnostic resources.

Kaplan-Meier analysis showed an overall death incidence rate of 0.85 per 100 person-days. Clear differences in survival were observed among demographic groups. Male patients had a higher death rate compared to females. Patients aged 50 and above experienced the highest death incidence at 2.71 per 100 person-days and the poorest survival

outcomes, with a 25th percentile survival time of just 8 days. This indicates that older adults carried a disproportionately high mortality burden. From the multivariable Cox regression model, age was a predictor of mortality, with individuals aged 50 years and above having substantially higher hazard of death compared to children aged 0–14 years. The increased risk in this age group may be linked to natural declines in immune function with age, as well as a greater prevalence of chronic conditions like hypertension and diabetes, which can worsen dehydration and metabolic imbalances caused by cholera, ultimately leading to slower recovery or higher chances of death [2,28].

Unexpectedly, the highest death rate was seen among individuals classified as "non-case/other," even though this group was relatively small. The Cox regression model revealed that the case classification variable was another significant determinant of survival. Patients categorized as "Suspect/Probable" and "Asymptomatic" had lower hazards of death compared to confirmed cases, suggesting that confirmed patients likely presented with more severe disease manifestations or delayed treatment. In contrast, individuals in the "Non-case/Other" category experienced significantly higher mortality hazards. This raises important concerns about the accuracy and consistency of case classification, especially under the pressures of an outbreak. This may reflect misclassification of severe cholera cases or patients suffering from other acute gastrointestinal illnesses that resemble cholera clinically but were not confirmed by laboratory testing due to limited diagnostic resources or delays [28]. Alternatively, some of these patients might have been in advanced stages of illness or had coexisting health conditions that increased their risk of death. This finding points to possible weaknesses in surveillance sensitivity, diagnostic accuracy, and triage procedures, emphasizing the urgent need to improve case classification, ensure timely and reliable diagnostics, and enhance data quality during outbreak responses [29].

In this study, the CFR among cholera patients was 4.2%, exceeding the national CFR of 3.5% reported as of February 2024 [22,23], and well above the World Health Organization's recommended threshold of 1% [30]. This high mortality rate points to ongoing challenges in cholera case management within Zambia, despite efforts under the National Multisectoral Cholera Elimination Plan (NCP) [19,20]. Previous research shows that without treatment, cholera mortality can surpass 50%, while prompt and appropriate care—especially oral and intravenous rehydration can lower the CFR to under 1% [2,21,31]. The elevated CFR found here suggests many patients may delay seeking care or receive treatment that is late or inadequate, leading to preventable deaths. This gap between policy goals and real-world outcomes likely reflects weaknesses in health system capacity, surveillance, and health-seeking behaviors [15,24,32]. Improving early case detection, streamlining referral pathways, and increasing access to timely treatment are essential steps to reduce cholera deaths in future outbreaks.

Further analysis by demographic and clinical subgroups revealed additional vulnerabilities. Patients aged 50 years and older were disproportionately impacted, while younger individuals especially those aged 0–14 accounted for nearly half of all cholera cases (48%). Although children are typically the most affected group during cholera outbreaks, their outcomes in this study were comparatively better. This may reflect earlier care-seeking, differences in disease severity, or other unmeasured factors related to age-related physiology or access to care [16,19,33,34]..

Sex-based differences were also apparent in this study. Males accounted for a greater proportion of cholera cases and experienced worse clinical outcomes than females and in the multivariable cox regression analysis sex emerged as a significant predictor, with males exhibiting nearly twice the hazard of death compared to females. This finding is consistent with evidence from previous cholera studies conducted in sub-Saharan Africa, which reported higher case fatality rates among males [35,36] (Morof et al, 2013; Gunnlaugsson, 2000). These observed disparities may result from factors such as occupational exposures, gender-related differences in health-seeking behavior, and possible biological variations in disease progression [19]. Additionally, cultural norms that discourage men from seeking care promptly highlight the need for gender-sensitive public health campaigns and targeted community outreach efforts [16].

The median time from symptom onset to death was just three days, and from admission to death only two days, suggests the urgent need for prompt and effective clinical care in cholera cases. This narrow timeframe aligns with longstanding knowledge of cholera's rapid progression, first documented during John Snow's 1854 investigation of the London

outbreak [19,37]. The dramatic difference between CFR over 50% without treatment versus under 1% with timely care [2,21] alongside the fact that many deaths occur outside health facilities [17,22], emphasizes both the life-saving impact of rapid treatment and the critical gaps in timely response. Although the median delay from symptom onset to first healthcare contact was zero days, the persistently high mortality suggests that simply presenting early is not enough [21,31]. This likely reflects issues with treatment quality or delays after arrival at health facilities [19,34]. The speed of fatal deterioration also challenges the relevance of the '7-1-7' timeliness benchmark a global target that calls for identifying suspected outbreaks within 7 days, reporting to authorities within 1 day, and mounting an effective response within 7 days because cholera can lead to death within hours, making a seven-day response window inadequate [23,32,38].

Geographical variation was evident, with Lusaka Province showing a markedly higher hazard of death compared to other provinces. Similar spatial disparities have been documented in prior outbreaks, where densely populated urban centers such as Lusaka experienced higher case fatality rates due to strained health infrastructure and delayed access to treatment centers [29,39]. This finding highlights the need for region-specific response strategies and improved emergency preparedness in high-risk urban areas.

## Strengths and weaknesses of the study

This study's strengths include its nationwide coverage and large sample size, which enhance both statistical power and the ability to generalize findings. By drawing on comprehensive surveillance data from urban and peri-urban treatment centers, the research offers detailed insight into care timelines across diverse demographic and clinical groups. However, there are several limitations to note. The study depends on routine surveillance data, which may be affected by misclassification and incomplete records. Cases with missing or inconsistent date information were excluded, potentially introducing selection bias. Additionally, some survival percentiles could not be estimated for certain subgroups due to right-censoring or small numbers of events, limiting the scope of comparisons. Moreover, while delays were quantified, the study did not explore qualitative factors influencing care-seeking behavior, such as perceptions of cholera or trust in the healthcare system, which are important for understanding the context behind observed delays.

## Implications of this study to clinical practice and research

The data indicates that a significant proportion of cholera deaths occur shortly after patient admission, this suggests the importance of early risk stratification and aggressive fluid management protocols upon arrival at treatment centres. Due to the noticeably higher CFR among older adults and males, triage protocols should take into account age- and sex-specific vulnerabilities. Additionally, the study reveals critical deficiencies in the utilization of diagnostics, underscoring an urgent need to expand point-of-care rapid diagnostic tests (RDTs) to facilitate swift case confirmation and tailored management strategies.

From a health systems perspective, the findings emphasize the necessity for community-based early warning systems and targeted health messaging aimed at high-risk groups. Future research should employ a mixed-methods approach to investigate the socio-cultural and infrastructural factors that contribute to delays in care and evaluate the effectiveness of real-time digital surveillance platforms in reducing response times. Furthermore, the integration of geospatial analyses could enhance the targeting of hotspots during future outbreaks.

## Recommendations

The findings of this study highlight several urgent areas for action to reduce cholera mortality and strengthen outbreak response in Zambia. Health facilities should prioritize rapid identification of high-risk patients, particularly older adults and males, who experienced the highest mortality in this outbreak. Strengthening triage protocols and ensuring that suspected cholera cases are managed promptly and systematically can help prevent deaths that occur within the first critical hours

of admission. The study also identified gaps in case classification and diagnostic confirmation, including high mortality among patients categorized as "non-case/other." To address this, rapid diagnostic tests should be expanded to facilitate timely confirmation of cases and guide appropriate care, while continuous training of healthcare workers in consistent application of case definitions can reduce misclassification and improve patient outcomes.

At the community and health system level, targeted public health messaging is needed to encourage early care-seeking among older adults and men, who may delay treatment or present with more severe illness. Engaging community leaders and health workers in culturally sensitive outreach can help overcome social or gender-based barriers to timely healthcare access. Surveillance systems should be strengthened to provide more accurate, real-time data, enabling health authorities to identify high-risk populations and respond quickly. Additionally, policy makers and health system planners should assess and address structural gaps, including staffing, supply chain management, and facility readiness, to ensure that resources are available where they are most needed during outbreaks. Finally, operational research should be supported to better understand barriers to timely care and to evaluate interventions aimed at reducing mortality. Implementing these measures, which are directly informed by the study's findings, can significantly enhance Zambia's cholera preparedness, response, and ultimately reduce preventable deaths.

## Acknowledgments

We gratefully acknowledge the Zambia National Public Health Institute (ZNPHI) and the Ministry of Health, Lusaka, Zambia, for their support and for providing access to the surveillance data used in this study.

## Author contributions

**Conceptualization:** Deborah Tembo, Miyanda Simwaka, Chipo Nkwemu, Nedah Chikonde Musonda, Samson Shumba.

**Data curation:** Deborah Tembo, Miyanda Simwaka, Chipo Nkwemu, Nedah Chikonde Musonda, Samson Shumba.

**Formal analysis:** Deborah Tembo, Miyanda Simwaka, Chipo Nkwemu, Nedah Chikonde Musonda, Samson Shumba.

**Methodology:** Deborah Tembo, Miyanda Simwaka, Chipo Nkwemu, Nedah Chikonde Musonda, Samson Shumba.

**Software:** Deborah Tembo, Miyanda Simwaka, Chipo Nkwemu, Nedah Chikonde Musonda, Samson Shumba.

**Validation:** Deborah Tembo, Miyanda Simwaka, Chipo Nkwemu, Nedah Chikonde Musonda, Samson Shumba.

**Visualization:** Deborah Tembo, Miyanda Simwaka, Chipo Nkwemu, Nedah Chikonde Musonda, Samson Shumba.

**Writing – original draft:** Deborah Tembo, Miyanda Simwaka, Chipo Nkwemu, Nedah Chikonde Musonda, Samson Shumba.

**Writing – review & editing:** Deborah Tembo, Miyanda Simwaka, Chipo Nkwemu, Nedah Chikonde Musonda, Samson Shumba.

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
