## [Decision Letter · Decision Letter 0]

16 Sep 2025

Dear Dr. Shumba,

Thank you for submitting your manuscript to PLOS ONE. After careful consideration, we feel that it has merit but does not fully meet PLOS ONE’s publication criteria as it currently stands. Therefore, we invite you to submit a revised version of the manuscript that addresses the points raised during the review process.

We look forward to receiving your revised manuscript.

Kind regards,

Olushayo Oluseun Olu

Academic Editor

PLOS ONE

Journal Requirements:

2. We note you have included a table to which you do not refer in the text of your manuscript. Please ensure that you refer to Table 2 and Table 5 in your text; if accepted, production will need this reference to link the reader to the Table.

5. Please include captions for your Supporting Information files at the end of your manuscript, and update any in-text citations to match accordingly. Please see our Supporting Information guidelines for more information: http://journals.plos.org/plosone/s/supporting-information .

Additional Editor Comments :

Please download the attached manuscript file which has the comments of reviewer 1 and ensure that you address all of them as well as those of reviewer 2. Kindly ensure that you provide point-by-point responses to all comments and indicate exactly (line and page numbers) where changes to the revised manuscripts were made. Thank you.

Reviewers' comments:

Reviewer's Responses to Questions

**Comments to the Author**

1. Is the manuscript technically sound, and do the data support the conclusions?

Reviewer #1: Yes

Reviewer #2: Yes

2. Has the statistical analysis been performed appropriately and rigorously?

Reviewer #1: Yes

Reviewer #2: No

3. Have the authors made all data underlying the findings in their manuscript fully available?

Reviewer #1: Yes

Reviewer #2: No

4. Is the manuscript presented in an intelligible fashion and written in standard English?

Reviewer #1: Yes

Reviewer #2: Yes

Reviewer #1: 6a. What figure was used for survival analysis denominator? (3655 or 2162?). This is confusing and needs to be reconciled bearing in mind that this may alter all subsequent calculations

6b. The exact dates or days of the months for which data was analyzed should be indicated

6c. Also cases up to December 2024 were said to have been included but August 2024 was indicated !!!!. There is a need for reconciliation

6d. What is meant by laboratory testing prior to notification?

6e. There should be a specific subsection for the study recommendations

6f. Other comments are in situ of the reviewed manuscript

Reviewer #2: The topic is important. However, there are several critical issues in study design, labelling, internal consistency, and others to be addressed before the paper is suitable for publication. I recommend major revisions.

Design

The study analyzes time-to-event outcomes with Kaplan–Meier curves and incidence rates. This is, in substance, a retrospective cohort (or surveillance cohort) study please correct.

All suspected and confirmed cholera cases reported between January 2023 and December 2024 were included,” but data were “accessed on 8 August 2024.” This implies inclusion of future records or retrospective completion beyond the access date. Please reconcile dates, state the last date of outcome ascertainment/closure.

Inconsistencies in counts and tables

Results state 3,655 patients total; 0–14 years = 37.8%. Later, Table 2 claims the 0–14 group “constituted the largest portion of the cohort (788 patients)” for survival, but another row for 0–14 appears to show “2,285” in the “Number of Patients” column ( which exceeds both the survival cohort (2,162) and contradicts earlier totals.

The survival table also lists percentiles (e.g., 25th percentile survival time of 16 days for males) that seem inconsistent with the narrative that deaths occur rapidly (median onset-to-death 3 days; admission-to-death 2 days), need to reconcile.

Others

Report 95% CIs for incidence rates (events per 100 person-days), log-rank tests, and any hazard ratios if you add regression.

Given that multiple risk factors are discussed (age, sex, case classification), a multivariable Cox model (or Poisson/negative binomial model with an offset for person-time) is needed to adjust for confounding.

You report RDTs performed in 11.2% of cases, but then state ELISA was “the most commonly used method” (76%). In cholera surveillance, culture and PCR are standard confirmatory tests; the “serological (ELISA)” method requires justification, specific assay names, and denominators.

The paper emphasizes “delays,” yet the descriptive statistic for symptom onset to first healthcare contact is median 0 days (IQR 0–2). Please present the full distribution (e.g., proportions arriving ≥24h, ≥48h) and stratify by outcome (died vs survived). If many deaths occur rapidly after arrival, the bottleneck may be in triage/initial resuscitation rather than pre-hospital delay.

Your statement that findings challenge the relevance of the 7-1-7 timeliness benchmark” lacks a supporting reference and seems beyond the scope of the presented data. Either substantiate with appropriate sources and a targeted analysis, or remove/soften this claim.

PLOS ONE requires that all data underlying results be available with rare exceptions. Your statement says ZNPHI/MoH holds data and are accessible upon request with permission. Please provide a contact mechanism (unit/email) and access criteria.

Use standard terms consistently (e.g., case fatality ratio [CFR] rather than “mortality rate” etc

Grammar/style: Several long sentences in Background/Discussion can be shortened and proof read again

**Do you want your identity to be public for this peer review?** For information about this choice, including consent withdrawal, please see our Privacy Policy

Reviewer #1: **Yes: ** Prof. Kayode OSAGBEMI

Reviewer #2: **Yes: ** Sylvester Maleghemi

---

## [Author Response · Author response to Decision Letter 1]

11 Oct 2025

Response to Reviewers

We would like to sincerely thank the Editor and the reviewers for overseeing this process and for providing constructive comments on our manuscript entitled "Rapid Progression, Delayed Response: Survival and Mortality Patterns in Zambia’s January 2023–July 2024 Cholera Outbreak." We have carefully addressed each concern raised and provide detailed responses below. We remain happy to make any further revisions as required.

Reviewer Comments and Responses

1. Comment: We note you have included a table to which you do not refer in the text of your manuscript. Please ensure that you refer to Table 2 and Table 5 in your text; if accepted, production will need this reference to link the reader to the Table.

Response: The comment has been addressed, see page 5, 8, and 10

2. Comment: Among the 2,162 patients included in the survival analysis (What figure was used for survival analysis? 3,655 or 2,162?). This is confusing and needs to be reconciled, as it may alter subsequent calculations.

Response: The issue has been resolved; see page 2. “Of the 3,655 cholera patients enrolled in the study, 2,162 patients were included in the survival analysis”

3. Comment: 13.9% were asymptomatic (How and why did asymptomatic patients get into treatment centers and included in any stage of analysis?).

Response: Although 13.9% of patients were asymptomatic, they were included because cholera surveillance often identifies contacts of confirmed cases or individuals presenting for testing during outbreaks. These patients may have sought care due to exposure concerns, mild symptoms, or routine screening. Including them allows the analysis to capture the full spectrum of reported cases and infection severity.

4. Comment: Cases up to December 2024 were said to have been included. This needs reconciliation.

Response: This was a typographical error and has been corrected; see page 5.

5. Comment: Patient admission status, vital status at reporting, and laboratory testing prior to notification what is this?

Response: The sentence has been clarified for improved flow and clarity; see page 5.

6. Comment: Males had an estimable 25th percentile survival time of 16 days, while all percentile estimates were unavailable for females. Why?

Response: Percentile survival times for females were not estimable because the number and timing of deaths were too few to reach the required proportion of events (25%, 50%, or 75%) needed for calculation.

7. Comment: Figure 4: Kaplan-Meier Survival Curves for Cholera Patients Stratified by Sex, Age, and Case Classification.

Response: Figures were generated in Stata and have been attached separately.

8. Comment: Although children are typically the most affected group, their outcomes were comparatively better. This may be due to targeted pediatric treatment protocols???

Response: The statement has been deleted; see page 13.

9. Comment: The findings stress the need for earlier triage, decentralized rehydration efforts (this does not emanate from the study), and targeted outreach for high-risk populations.

Response: The statement has been deleted; see page 13.

10. Comment: Results state 3,655 patients total; 0–14 years = 37.8%. Later, Table 2 claims the 0–14 group “constituted the largest portion of the cohort (788 patients)” for survival, but another row for 0–14 appears to show “2,285” in the “Number of Patients” column ( which exceeds both the survival cohort (2,162) and contradicts earlier totals.

Response: It was a typo it has been corrected see page 9

11. Comment: Report 95% CIs for incidence rates (events per 100 person-days), log-rank tests, and any hazard ratios if you add regression.

Given that multiple risk factors are discussed (age, sex, case classification), a multivariable Cox model (or Poisson/negative binomial model with an offset for person-time) is needed to adjust for confounding.

Response: The log-rank tests were included on the Kaplan Meier just the next section. The Multivariable Cox regression has been included.

12. Reviewer: You report RDTs performed in 11.2% of cases, but then state ELISA was “the most commonly used method” (76%). In cholera surveillance, culture and PCR are standard confirmatory tests; the “serological (ELISA)” method requires justification, specific assay names, and denominators.

Response: Although culture and PCR are the standard confirmatory methods for cholera, ELISA-based serology was also used during the outbreak as part of the national surveillance protocol. ELISA tests (LOINC 44012-3) were primarily applied in settings where culture or PCR capacity was limited, providing rapid preliminary confirmation of suspected cases. Including ELISA allowed broader coverage and timely reporting in resource-constrained settings. The manuscript now specifies the assay names, LOINC codes, and denominators for all test types to ensure clarity and reproducibility.

13. Reviewer: The paper emphasizes “delays,” yet the descriptive statistic for symptom onset to first healthcare contact is median 0 days (IQR 0–2). Please present the full distribution (e.g., proportions arriving ≥24h, ≥48h) and stratify by outcome (died vs survived). If many deaths occur rapidly after arrival, the bottleneck may be in triage/initial resuscitation rather than pre-hospital delay.

14. Comment: This may be due to targeted pediatric treatment protocols, earlier healthcare seeking, or greater physiological resilience among younger patients.

Response: Wording has been revised to reflect multiple plausible factors without speculation: improved pediatric outcomes likely result from specialized treatment protocols, prompt care-seeking, and potential physiological resilience; see page 13

15. Comment: The speed of fatal deterioration also challenges the “7-1-7” timeliness benchmark. Please explain.

Response: We have clarified that “7-1-7” refers to detecting outbreaks within 7 days, notifying authorities within 1 day, and initiating an effective response within 7 days. Given cholera’s rapid progression, this benchmark may be insufficient; see page 14.

16. Comment: Community-based early warning systems and ORPs do not emanate from this study.

Response: The statement has been deleted; see page 15.

17. Comment: Recommendations: There is a critical need to include a subsection with actionable, stakeholder-targeted recommendations.

Response: We thank the reviewer for this important point. A new ‘Recommendations’ subsection has been added to the Discussion, providing actionable guidance for stakeholders; see page 16.

18. Comment: Please confirm whether there are legal or ethical restrictions on sharing your data publicly.

If legal or ethical restrictions apply, please provide all necessary instructions and non-author contact information (preferably email) for a data access committee, ethics committee, or other institutional body that other researchers would require to request access to your data.

Response: Dataset has been attached.

Closing:

We sincerely appreciate the reviewers’ careful reading of our manuscript and their constructive feedback, which has substantially improved the clarity and quality of the paper. We hope that the revisions satisfactorily address all comments.

---

## [Decision Letter · Decision Letter 1]

25 Nov 2025

Rapid Progression and Short-Term Mortality during the January 2023 to July 2024 Cholera Outbreaks in Zambia: A Retrospective Facility-Based Study

PONE-D-25-43434R1

Dear Dr. Shumba,

We’re pleased to inform you that your manuscript has been judged scientifically suitable for publication and will be formally accepted for publication once it meets all outstanding technical requirements.

Kind regards,

Olushayo Oluseun Olu

Academic Editor

PLOS ONE

Additional Editor Comments (optional):

Reviewers' comments:

Reviewer's Responses to Questions

**Comments to the Author**

Reviewer #1: All comments have been addressed

2. Is the manuscript technically sound, and do the data support the conclusions?

Reviewer #1: Yes

3. Has the statistical analysis been performed appropriately and rigorously?

Reviewer #1: Yes

4. Have the authors made all data underlying the findings in their manuscript fully available?

Reviewer #1: Yes

5. Is the manuscript presented in an intelligible fashion and written in standard English?

Reviewer #1: Yes

Reviewer #1: The issues raised in previous review have been adequately corrected. The Journal may go ahead and publish the article as attached.

**Do you want your identity to be public for this peer review?** For information about this choice, including consent withdrawal, please see our Privacy Policy

Reviewer #1: No

---

## [Editor Report · Acceptance letter]

PONE-D-25-43434R1

PLOS ONE

Dear Dr. Shumba,

I'm pleased to inform you that your manuscript has been deemed suitable for publication in PLOS ONE. Congratulations! Your manuscript is now being handed over to our production team.

Kind regards,

on behalf of

Dr. Olushayo Oluseun Olu

Academic Editor

PLOS ONE